

# Shipborne measurements of Antarctic submicron organic aerosols: an NMR perspective linking multiple sources and bioregions

Stefano Decesari[1*], Marco Paglione[1]; Matteo Rinaldi[1]; Manuel Dall'Osto[2], Rafel Simó[2]; Nicola Zanca[1]; Francesca Volpi [1]; Maria Cristina Facchini[1]; Thorsten Hoffmann[3]; Sven Götz[3]; Christopher Johannes Kampf[4]; Colin O'Dowd[5]; Jurgita Ovadnevaite[5]; Darius Ceburnis[5]; Emilio Tagliavini[6].

[1]Institute of Atmospheric and Climate Sciences, National Research Council of Italy (CNR), I-40129, Bologna, Italy.

[2]Institut de Ciències del Mar, Consejo Superior de Investigaciones Científicas (CSIC), ES-08003, Barcelona, Spain.

[3]Institute of Inorganic and Analytical Chemistry, Johannes Gutenberg University of Mainz, 55128, Mainz, Germany.

[4]Institute of Organic Chemistry, Johannes Gutenberg University of Mainz, 55128, Mainz, Germany.

[5]School of Physics and C-CAPS, National University of Ireland Galway, H91 CF50, Galway, Ireland.

[6]Department of Chemistry, University of Bologna, 40126, Bologna, Italy.

*corresponding author.

Keywords: Antarctic aerosols, natural sources of aerosol in polar regions, marginal ice zones, organic aerosol, NMR spectroscopy, primary marine particles, secondary organic aerosol, atmospheric amines

**Abstract**
The concentrations of submicron aerosol particles in maritime regions around Antarctica are influenced by
the extent of sea ice. This effect is two way: on one side, sea ice regulates the production of particles by sea
spray (primary aerosols) while, on the other side, it hosts complex communities of organisms emitting
precursors for secondary particles. Past studies documenting the chemical composition of fine aerosols in
Antarctica indicate various potential primary and secondary sources active in coastal areas, in offshore
marine regions as well as in the sea ice itself. In particular, beside the well-known sources of organic and
sulfur material originating from the oxidation of dimethyl-sulfide (DMS) produced by microalgae, recent
findings obtained during the 2015 PEGASO cruise suggest that nitrogen-containing organic compounds are
also produced by the microbiota colonizing the marginal ice zone. To complement the aerosol source
apportionment performed using online mass spectrometric techniques, here we discuss the outcomes of
offline spectroscopic analysis performed by nuclear magnetic resonance (NMR) spectroscopy. In this study
we (i) present the composition of ambient aerosols over open ocean waters across bioregions, and
compared it to the composition of (ii) seawater samples and (iii) bubble bursting aerosols produced in a sea
spray chamber on board the ship. Our results show that the process of aerosolization in the tank enriches
primary marine particles with lipids and sugars while depleting them of free aminoacids, providing an
explanation for why aminoacids occurred only at trace concentrations in the marine aerosol samples



analyzed. The analysis of water-soluble organic carbon (WSOC) in ambient submicron aerosol samples
shows distinct NMR fingerprints for three bioregions: 1) the open Southern Ocean pelagic environments, in
which aerosols are enriched with primary marine particles containing lipids and sugars; 2) sympagic areas in
the Weddell Sea where secondary organic compounds, including methanesulfonic acid and semivolatile
amines abound in the aerosol composition; and 3) terrestrial coastal areas, traced by sugars such as
sucrose, emitted by land vegetation. Finally, a new biogenic chemical marker, creatinine, was identified in
the samples from the Weddell Sea, providing another confirmation of the importance of nitrogen-
containing metabolites in Antarctic polar aerosols.

## 1. Introduction

The Antarctic continent is one of the last pristine areas of our planet but its natural ecosystems are now
threatened by an acceleration of the effects of global warming. Although at the beginning of the XXI
centuries the signals of climate change looked still weak in the region, the ice-sheet mass loss in Western
Antarctica has greatly accelerated in the last ten years as the Southern Ocean waters have experienced a
clear warming trend (Shepherd et al., 2018). The consequences for Antarctic maritime and coastal
environments encompass strengthening of westerly winds, reduction of summer sea ice extent, shifting
geographical ranges of bird communities, expanding terrestrial vegetation, increasing glacier melt and
freshwater formation over land, etc. (Rintoul et al.; 2018). As all these specific ecosystem impacts involve
factors deemed important for aerosol production in Antarctica (Davison et al., 2006; Schmale et al., 2013;
Kyrö et al., 2013; Barbaro et al. 2017), a significant effect of climate change on atmospheric concentrations
of aerosols and cloud condensation nuclei (CCN) must be expected to occur by the end of this century. The
field studies performed in maritime and coastal areas around Antarctica in the austral summer since the
90's (Davison et al., 1996) enable us to gather precious information on the multiple feedbacks between
atmospheric composition and ecosystems in a warming climate. In summer, the sea ice recedes allowing
wind stress over the oceanic surface and sea spray occur closer to the continent, hence increasing the
production of primary marine aerosols. At the same time, the thinning of sea ice in its marginal zone and
the increased intensity of solar radiation allow microalgae colonize the ice (Fryxell and Kendrick, 1988;
Roukaerts et al. 2016). The microbiota produce low-molecular weight metabolites as cryoprotectors and
osmoregulators, like dimethylsulfoniopropionate (DMSP) and quaternary nitrogen compounds (Dallosto et
al., 2017). Once released in seawater, such compounds become precursors of atmospheric reactive volatile
reactive compounds, such as dimethylsulfide (DMS) and methylamines, which eventually can lead to the
formation of secondary aerosols. Indeed, Davison et al. (1996) observed concentrations of DMS south of
60°S more than four times higher than in the Atlantic Ocean.
During the 2015 PEGASO cruise (Dall'Osto et al., 2017, 2019; Fossum et al., 2018), we conducted
continuous atmospheric observations for over 42 days, providing one of the longest shipborne aerosol
measurement records in this area of the world. We contrasted the composition of seawater north and
south of the Southern Boundary of the Antarctic Circumpolar Current (SBACC), which represents the
approximate boundary between the open Southern Ocean and the waters directly affected by sea ice
formation and melt around Antarctica. Dall'Osto et al. (2017) showed that not only DMSP and DMS
occurred in greater concentrations in sympagic waters (south of the SBACC), but so did quaternary nitrogen
compounds and methyl amines. By contrast, other biological parameters of seawater, like chlorophyll $a$,
total organic carbon (TOC) and transparent exopolymeric particles (TEP), showed higher concentrations
north of the SBACC, in the open Southern Ocean. These results indicate that not only the biological
productivity per se, but also the composition of the microbiota affect the production of aerosol precursors
in seawater. Indeed, the observations of organic nitrogen in the aerosol – carried out by both online and





offline chemical methods – pointed to strong sources in the area of the Weddell Sea which, at the time of
the field campaign, was heavily covered by sea ice.
These findings contribute to the growing observational dataset of aerosol chemical compositions for
coastal Antarctic and sub-Antarctic marine areas, which hosts reports of chemical analysis performed on
filter and impactor samples (Davison et al., 2006; Virkkula et al., 2006) as well as the results of online
aerosol mass spectrometric techniques acquired in the recent years (Zorn et al., 2008; Schmale et al., 2013;
Giordano et al., 2017). All the measurements performed so far agree in showing a reduction of sea salt
aerosols from the Southern Ocean to the coasts of Antarctica, while secondary species including non-sea
salt sulfate and methanesulfonate (MSA) occur in relatively higher concentrations at higher latitudes as a
result of the DMS emissions from marginal ice zone waters. Open questions remain about a) the amount of
non-MSA organic matter in Antarctic air masses, and b) its origin (either primary or secondary). Recent
studies also suggests that blowing snow at high wind speeds may be an important yet hitherto
underestimated source (Giordano et al, 2018; Frey et al., 2019), adding complexity onto the source
apportioning of organic aerosols. First observations of organic carbon (OC) in size-segregated aerosol
samples collected at a coastal site in the Weddell Sea (Virkkula et al., 2006) showed that MSA represented
only a few % of the total OC in the submicron fraction. In contrast with these findings, aerosol mass
spectrometric (AMS) measurements showed that the organic matter in submicron aerosols transported in
Antarctic air masses was almost totally accounted for by MSA, while non-MSA organic compounds were
associated to aerosols originating from highly productive waters in the Southern Ocean (Zorn et al., 2008).
Non-MSA OC can form also from insular terrestrial biomass emissions (Schmale et al., 2013). In particular,
organic particles emitted from seabird colonies contain large amounts of nitrogen with MS spectral
fingerprints overlapping with those of natural aminoacids. In the paper by Liu et al. (2018), FTIR
spectroscopy was employed to probe the sources of particulate organic compounds at another coastal
Antarctic site, and the results point to a contribution of marine polysaccharides transported in sea spray
aerosols. Finally, detailed organic speciation using offline analytical techniques with high sensitivity and
selectivity suggest further contributions from marine proteinaceous material, terrestrial lipids, and
secondary organic compounds (Bendle et al 2007; Barbaro et al., 2015; 2017), but it is unclear how much
the concentrations of compounds occurring at pg m$^{-3}$ relate to that of bulk organic matter. We present here
the organic characterization of Antarctic aerosol employing proton nuclear magnetic resonance ($^1$H-NMR)
spectroscopy. NMR spectroscopy has been used for decades in several fields of biogeochemistry for its
ability to fingerprint several classes of biomolecules and natural organic matter in aquatic and terrestrial
environments (e.g., Pautler et al., 2012; Hertkorn et al., 2013). In this study, which focuses on the analysis
of samples collected during the PEGASO 2015 cruise, we contrast the NMR composition of submicron
aerosol samples with that of seawater samples and bubble-bursting aerosols. The results provide new hints
on the origin of non-MSA aerosol organic matter in fine aerosol particles in the Antarctic and sub-Antarctic
marine environment.

**2. Experimental**
**2.1. Ambient aerosol sampling on filters.**
The PEGASO (Plankton-derived Emissions of trace Gases and Aerosols in the Southern Ocean) cruise was
conducted on board of *RV Hesperides* in the regions of Antarctic Peninsula, South Orkney and South
Georgia Islands from 2 January to 11 February 2015 (Dall'Osto et al. 2017). A high volume sampler (TECORA
ECO-HIVOL, equipped with Digitel PM1 sampling inlet) collected ambient aerosol particles with Dp < 1 µm
on pre-washed and pre-baked quartz-fibre filters, at a controlled flow of 500 LPM. Sampling was allowed
only when the samplers were upwind the ship exhaust with a relative wind speed threshold of 5 m s$^{-1}$. Due





to the necessity of collecting sufficient amounts of samples for detailed chemical analyses, sampling time
was of the order of ~50 h for each sample. A total of eight PM1 samples were collected during the cruise
(Figure 1). The samples were stored at −20 °C until extraction and NMR analysis.
For HPLC-MS analyses, aerosol samples were collected on PTFE fiber filters (70 mm diameter, Pallflex
T60A20, Pall Life Science) with flow rates of 2.31 – 2.41 $m^3$/h through a PM2.5 inlet. Sampling times ranged
from 12 - 24 h, resulting sampling volumes of 28.1 – 56.1 $m^3$ of air. As outlined above, sampling was only
allowed when the sampler was upwind the ship exhaust.
**2.2. Seawater sampling and tank experiments.**
Seawater samples were collected from a depth of 4 m using either the uppermost Niskin bottle of the CTD
rosette casts or the ship's flow-through underway pumping system. The samples were filtered with a
Millipore filtration apparatus on quartz-fiber filters (Whatman, Ø= 47mm) after a previous cut off at 10 μm
performed with a polycarbonate filter (Millipore, Isopore, porosity=10 μm, Ø= 47mm). In total 45 samples
were collected for subsequent quantification of the Particulate Organic Carbon (POC) and 20 mL of the
filtrates were stored for subsequent analysis of Dissolved Organic Carbon (DOC). All the samples were
stored at −20 °C until the chemical analyses. Three samples of sea ice from the marginal ice zone in the
northern Weddell Sea were also collected using the methodology described in Dall'Osto et al. (2017). The
samples, once melted, were filtered and treated similarly to the seawater samples.
Seawater was pumped from a depth of 4 m to fill an airtight high grade stainless steel tank (200 L) designed
for aerosol generation experiment. Sea ice samples were also introduced and melted in the tank for
dedicated experiments. Water was dropped from the top of the tank as a plunging jet at a flow rate of 20 L
$min^{-1}$. The entrained air formed bubbles that, upon bursting, produced sea-spray aerosol, as reported in
O'Dowd et al. (2015). Particle-free compressed air was blown into the tank headspace (120 L $min^{-1}$), which
had outlet ports leading to samplers for the collection of filters and the subsequent off-line chemical
characterization of the produced sea-spray. In particular nine sea-spray aerosol samples were collected for
approximately 72h by a PM1 sampler (flow rate 40 lpm) equipped with pre-washed and pre-baked quartz-
fiber filters (PALL, Ø= 47mm). Sampling time was of the order of ~72 h for each sample.
**2.3. $^1$H-NMR spectroscopy.**
Quartz-fiber filters from both ambient, POC filter samples and sea-spray generation experiments were
extracted with deionized ultra-pure water (Milli-Q) in a mechanical shaker for 1 h and the water extract was
filtered on PTFE membranes (pore size: 0.45 μm) in order to remove suspended particles. The water-
soluble organic carbon (WSOC) content was quantified using a TOC-TN thermal combustion analyser (Multi
N/C 2100 by Analytik Jena) (Rinaldi et al., 2007). Aliquots of the aerosol extract were dried under vacuum
and re-dissolved in deuterium oxide ($D_2O$) for organic functional group characterization by $^1$H-NMR
spectroscopy, as described in Decesari et al. (2000). The $^1$H-NMR spectra were acquired at 600 MHz in a 5
mm probe using a Varian Unity INOVA spectrometer, at the NMR facility of the Department of Industrial
Chemistry (University of Bologna). Sodium 3-trimethylsilyl-(2,2,3,3-$d_4$) propionate (TSP-$d_4$) was used as an
internal standard by adding 50 μL of a 0.05% TSP-$d_4$ (by weight) in $D_2O$ to the standard in the probe. To
avoid the shifting of pH-sensitive signals, the extracts were buffered to pH ~ 3 using a deuterated-
formate/formic-acid ($DCOO^-$/HCOOH) buffer prior to the analysis. The speciation of hydrogen atoms bound
to carbon atoms can be provided by $^1$H-NMR spectroscopy in protic solvents. On the basis of the range of
frequency shifts, the signals can be attributed to H-C containing specific functionalities (Decesari et al.,
2000, 2007). A total of eight HiVol PM1 ambient aerosol samples (+ one blank), four POC samples from
seawater and two POC samples from melted sea ice, and two samples from the tank experiments (from
aerosolization of one seawater sample and one melted sea ice sample) + one blank for the 47mm filters
were characterized by NMR spectroscopy.




**2.5. UHPLC-HESI-Orbitrap-MS.**

One half of each filter sample was extracted according to the following protocol: three times sonication in 1.5 mL, 1 mL, and 1 mL ACN/H2O (9:1, v/v) for 30 min. The extracts were filtered through PTFE membranes (pore size: 0.45 μm), combined, dried at 50 °C under a gentle stream of $N_2$, resuspended in 200 μL ACN/H2O (1:4, v/v), and stored at -20 °C until analysis. Samples were analyzed in triplicate by UHPLC-HESI-HRMS using an Orbitrap mass analyzer (Q-Exactive hybrid quadrupole orbitrap mass spectrometer, Thermo Scientific, Germany) equipped with an UHPLC-System (Dionex UltiMate 3000 UHPLC system, Thermo Scientific, Germany) and a Hypersil Gold, C18, 50 x 2.0 mm column with 1.9 μm particle size (Thermo Scientific, Germany). The injection volume was 20 μL and the eluents were ultrapure water with 2% acetonitrile and 0.04% formic acid (eluent A), and acetonitrile with 2% water (eluent B). The gradient of the mobile phase with a flowrate of 0.5 mL min-1 was as follows: starting with 2% B isocratic for 1 min, increasing to 20% B in 0.5 min, isocratic for 2 min, increasing to 90% B in 2.5 min, isocratic for 4 min and decreasing to 2% B in 0.5 min. Mass spectrometric analyses were performed using a ESI source under the following conditions: 30°C ESI temperature, 4 kV spray voltage, 40 psi sheath gas flow, 20 psi auxiliary gas flow and 350°C capillary temperature. Mass resolution was 70000 and the acquired mass range was m/z 80–550.

172

**2.5. Air mass back-trajectories.**

Five-day back trajectories arriving at the ship's position at 03:00, 09:00, 16:00 and 21:00 every day were calculated using the HYSPLIT model (Draxler & Rolph, 2010) with GDAS data (or the BADC Trajectory Service?). In total, 140 air mass back trajectories were obtained. A Polar Stereographic map was used to classify 24x24 km grid cells as land, sea and ice. From this information we calculated the percentage of time spent by each trajectory over each surface type, and particularly over sea ice. Daily maps of sea ice percentage concentration measured on a 12.5 km grid were used for this calculation. Sea ice abundance was derived from satellite microwave data (Ezraty et al., 2007) available at IFREMER. This analysis allowed also assigning air mass trajectories (and percentages of surface type overflown) to the aerosol samples collected on the filters (Figure 1).

183

**3. Results**

**3.1. Organic composition of seawater: POC samples.**

The composition of seawater in terms of pigments, metabolites, fluorescent organic matter and other organic constituents from the PEGASO cruise has been characterized in great detail (Dall'Osto et al., 2017; Nunes et al., 2019; Zamanillo et al., 2019). Marine organic substances occur in the ocean in dissolved and particulate form. Particulate organic carbon (POC) is defined operationally by a filtration cutoff at 0.45 μm, and recovers phytoplankton cells, bacteria and of the large colloids, such as transparent exopolymeric particles ("TEPs") (Passow et al., 2002). Dissolved organic carbon (DOC) is mostly contributed by the excreta and metabolites of the marine biota but it also accounts for a pool of refractory compounds, resistant to microbial degradation, and well mixed in the water column (Hertkorn et al., 2013). Past studies have extensively characterized the NMR features of labile and refractory organic constituents of marine organic matter (Repeta 2015). However, the NMR characterization of the dissolved organic substances was limited to desalted fractions of DOC isolated by solid-phase extraction or ultrafiltration (Koprivnjak et al., 2009). Therefore, the NMR analysis of low-molecular weight polar organic constituents of marine DOC remains elusive. In our study, we screened the NMR features of POC in phytoplankton bloom areas. In addition, samples of aerosolized seawater and melted sea ice were used as a proxy of primary marine aerosol





(Dall'Osto et al., 2017). During the process of bubble bursting performed in the tank experiments, aerosol
particles became depleted in sea salt with respect to seawater and enriched in surface-active DOC
components and in buoyant POC substances.
Figure 2 shows the proton NMR spectra of three POC samples, one from seawater (POC W3101) and two
from melted sea ice (POC SeaIce-1, and POC SeaIce-3) as examples. It is worth reminding that the samples
were pre-filtered through a polycarbonate membrane of 10 μm porosity, hence the analyzed POC fraction
is only the fine one (between ~ 0.45 and 10 μm). The interpretation of the spectra was carried out by
comparison with the datasets and spectra provided by the literature on metabolomics (e.g., Bertram et al.,
2009; Matulova et al., 2014; Li et al., 2015; Upadhyay et al., 2016) as well as by means of NMR analysis of
commercial standard compounds. Characteristic patterns of NMR resonances for specific compounds (e.g.,
patterns in multiplicity) enabled an accurate identification, while only a tentative attribution of the most
simple NMR resonances (singlets) was attempted when standards were not available, because deviations
with respect to published NMR data are possible when different experimental conditions (e.g., in respect to
pH of the sample) are used. Nevertheless, the POC extracts show several NMR features overlapping with
typical ones for other biological matrices. In particular, the occurrence of most common aliphatic
aminoacids was observed in all three samples analysed and particularly in sample POC SeaIce-1. Acidic
aminoacids dominated over the basic ones, while aromatic residuals were detected only in trace amounts
(Figure S1). The identification of modified aminoacids among the most typical natural products of the
Antarctic microbiota, such as mycosporines (Oyamada et al., 2007), could not be carried out in detail
because of the lack of suitable spectral libraries. The presence of metabolites such as low-molecular weight
nitrogen-containing compounds (choline, betaine, etc.) is confirmed by the singlets in the chemical shift
range 3.1 – 3.3 ppm from methyls bound to nitrogen atoms ($H_3C$-N-). Resonances at higher chemical shift,
between 3.4 and 4.2, recovered the -NC$\underline{H}$RCO- groups of alpha-aminoacids and the H-C-O groups of sugars
and polyols: traces of glycerol were found in all three samples analyzed, while glucose was found in trace
amounts in samples POC W3101 and as a major component in sample POC SeaIce-3 (Figure S2). These
results confirm the potential of [1]H-NMR spectroscopy for the characterization of marine metabolites and
natural products. The small set of POC samples analyzed in this study is, however, mainly aimed to provide
spectral fingerprints useful for the interpretation of the results of the aerosol sample analyses discussed in
the following sections.
**3.2. Organic composition of bubble bursting aerosols.**
The natural process of sea spray – mimicked by the experiments carried out in the tank onboard *RV
Hesperides*– selectively transfers organic compounds from seawater into the aerosol depending on the
ability of the specific pools of organic substances to enrich in the surface microlayer and/or to be
scavenged by rising air bubbles. The selective nature of such process is witnessed by our NMR data,
showing that the seawater composition dominated by aminoacids, osmolytes and sugars/polyols differs
quite substantially from that of bubble bursting aerosols from the tank experiments (Figure 3, Figure S3).
Bubble bursting aerosol was characterized by the occurrence of low-molecular weight metabolites like
lactic acid and amines (dimethylamine, DMA and traces of monomethyl- and trimethyl- amines) which likely
originated from DOC components of seawater. The most characteristic feature of the spectra is, however,
the bands at 0.9 and 1.3 ppm of chemical shift. These correspond to aliphatic chains with terminal methyl
moyeties typical of lipids. Their occurrence in the aerosolized seawater and not in the POC samples can be
explained by an enrichment of surface-active compounds from DOC in the surface microlayer. Lipid
enrichment in aerosol from bubble bursting experiments has already been documented by the two
previous studies reporting NMR composition data (Facchini et al., 2008; Schmitt-Kopplin et al., 2012).
Nevertheless, our findings clearly show that, beside lipids, there are specific constituents of POC taking part
in the formation of primary aerosol particles in the tank experiments. In particular, the spectral region for





sugars and polyols in bubble bursting aerosols is completely consistent with the spectral features of POC
(Figure S4), although the contribution of the -NCHRCO- groups of aminoacids in the same spectral window
is clearly missing in the aerosol. The presence of nitrogen-containing metabolytes (betaine) is confirmed in
the aerosol samples from the tank. It is plausible that betaine, glycerol and other sugars are chemically
bond to lipids making glycolipids and phospholipids, which can explain their preferential enrichment during
the aerosolization process with respect to other POC constituents like the aminoacids. It is a matter of fact
that aminoacids could be detected only in very trace amounts (the doublet of alanine at 1.45 ppm is barely
visible) in the sea spray samples. Other molecular tracers found in previous sea-spray experiments in other
geographical regions, such as acrylic acid (Schmitt-Kopplin et al., 2012), which is also product of DMPS
degradation, were not found in our experiment.

### 3.3. Organic composition of ambient submicron WSOC samples.

The eight ambient PM1 HiVol samples analyzed for organic composition include six that were collected in
parallel to the impactor samples discussed in Dall'Osto et al. (2017). The proton NMR spectra of the eight
samples are reported in Figures S5-S7. Air mass origin varied largely during the cruise, with transport from
the Weddell sea prevalent during the first half of the cruise turning into open ocean prevailing air masses
during the second half (Fig. 1). Two samples (A-0701 and A-0102) of mixed origin had been omitted by
Dall'Osto et al (2017), who focused on the comparison between aerosols from the sympagic regions and
those from the open ocean. We applied hierarchical cluster analysis to investigate if a dual classification
also held with the NMR spectra (Figure 4). The original spectra were normalized to their integrals and
binned to 354 points before clustering. Two main clusters were indeed identified: a first one recovering
three samples collected downwind the Weddell Sea during the first half of the cruise, and a second cluster
with samples representative of a greater diversity of conditions, from the Drake Channel, to the Antarctic
Peninsula and to the productive waters around South Georgia. This second cluster corresponds to the
samples characteristic for the open ocean conditions in Dall'Osto et al. (2017) plus samples A-0701 and A-
0102. Unexpectedly, sample A-0701, whose air mass spent most of time over sympagic waters (Fig. 1)
clustered together with the samples from the open ocean according to NMR composition. It is noticeable,
however, that binned NMR spectra can only trace the distribution of the major organic functional groups
while the information carried by fine spectral features, which is critical to detect the presence of specific
molecular markers, is not taken into account in the cluster analysis. In the following sections, we will show
that sample A-0701 exhibits a peculiar NMR composition which must be put in relation to terrestrial
sources of organic compounds. On the basis of the back-trajectories (Figs. 1 and 8), the likely land sources
were located in the Antarctic Peninsula. In summary, the variability in the distribution of NMR functional
groups in ambient PM1 samples (Table 1) was primarily driven by the air mass origin over sympagic
(Weddell Sea) or pelagic waters, in agreement with the results on inorganic compounds, WSOC and amines
reported by Dall'Osto et al. (2017; 2019). Nevertheless, the analysis of fine NMR spectral features supports
the existence of a third source area over land. In the following discussion, we will provide an in-depth
description of the NMR compositions for these three source sectors.

### 3.3.1. Ambient aerosols from the Weddell Sea.

Sample A-0911 was collected in the marginal ice zone of the Weddell Sea. Its spectrum shows a complete
absence of aromatic compounds and alkenes (Figure S7). The aliphatic region (Figures 5, S8) exhibits broad
similarity to that of the primary marine particles generated in the sea spray tank, but with a major
difference in the chemical shift range between 1.7 and 3.0 ppm where the background broad NMR bands
are much more intense in the ambient sample. This is also the region recovering the signals from acyl
groups (RCH-(C=O)-) in aliphatic carboxylic acids and ketoacids, which are formed by VOC oxidation in the
atmosphere (Barbaro et al., 2017). The most abundant individual compounds detected in these samples



were, however, MSA (Fossum et al., 2018), and the low-molecular methylamines (MMA, DMA, TMA). The
predominance of semivolatile $C_1$-$C_3$ alkyl-amines (Ge et al., 2011) indicates that the amines form in the
ambient aerosol by secondary processes involving volatilization from the ocean surface and recondensation
onto acidic aerosol particles (Dall'Osto et al., 2019). The aliphatic bands at 0.9 and 1.3 ppm in sample A-
0911 show a partial overlap with the resonances of the lipids in the aerosolized seawater. However, the
bands at 1.6 ppm and 2.2-2.3 ppm which, in lipids, correspond to methylenes in beta and alpha position to
a C=O group, are much more intense in the spectrum of A-0911 than in BB SeaIce-3 (Figure S8), indicating
that aliphatic chains are shorter and more substituted in the ambient aerosol than in nascent primary
aerosol particles. The pattern of bands at 0.9, 1.3, 1.6, 2.2, 2.4 and 2.6 ppm follow the structure elucidated
by Suzuki et al. (2001) and attributed to $C_7$-$C_9$ aliphatic dicarboxylic acids and oxo-acids. This class of organic
compounds, clearly characterizing the aliphatic composition of the ambient samples in the Weddell Sea
area, can originate from degraded (oxidized) lipids (Kawamura et al., 1996), or from gas-to-particle
conversion of carbonyls produced by the photochemical oxidation of lipids at the air-sea interface (Bernard
et al., 2016; Alpert et al., 2017). Support to the latter hypothesis (secondary formation) is given by the fact
that the N-osmolytes (betaine, choline) present in the sea spray generated in the tanks were completely
absent in the ambient sample. Nevertheless, the resonances in the spectral window 3.5 – 3.8 ppm in
sample A-0911 are completely consistent with the occurrence of glycerol, indicating that in fact primary
aerosol particles contributed to the composition of the ambient aerosol in this region (Figure S9). There is
another one striking difference between the composition of the ambient aerosol and sea spray particles:
the former contains significant levels (1.65 ng m$^{-3}$) of creatinine. This compound is responsible to the two
singlets at 3.12 ppm and 4.27 ppm of chemical shift and was identified by the comparison with a standard
under identical NMR experimental conditions (Figure S11). The concentration of creatinine clearly follows
that of low-molecular weight amines (Figure 6) and shows a maximum in the three samples collecting most
of the air masses that travelled over the Weddell Sea. Creatinine was also determined by HPLC/-MS analysis
in a parallel set of filter samples collected onboard Hesperides during the PEGASO cruise (see Section 2.4).
Identification was based on MS/MS fragmentation patterns and retention time. Quantification was based
on chromatographic peak area. Figure 7 shows extracted ion chromatograms for m/z 114.0655-114.0667,
corresponding to creatinine, of the filter extract of sample 0119N obtained during the PEGASO campaign
and the neat creatinine standard. The HPLC/MS analysis indicate that creatinine occurred in concentrations
of 20 – 50 pg/m$^3$ in the samples from the Weddell Sea area (Table S1), much less than the concentrations
determined by $^1$H-NMR spectroscopy (1.6 – 2.5 ng/m$^3$, Table 1). Such discrepancy can be due to the
different extraction protocols and to non-ideal chromatographic conditions in HPLC/MS for creatinine
quantification (elution close to the void volume). Nevertheless, our findings demonstrate that high-field
NMR methods can integrate HPLC/MS analysis for the identification of molecular markers in atmospheric
aerosol complex organic mixtures.
**3.3.2. Ambient aerosols in the open ocean.**
Sample A-2401 was collected during the northern transit of the cruise, *RV Hesperides* just west to South
Georgia (55° S) (Figure 1). During sampling, the air masses had a westerly component and can be
considered representative of Southern Ocean conditions. The $^1$H-NMR spectrum of A-2401 shares
similarities with that of A-0901 described above: a) the resonances of MSA and methyl-amines are much
more intense than that of other low-molecular weight compounds (such as N-osmolytes); b) the spectral
region of acyls (1.8 – 3.0 ppm) accounting for unresolved carboxylic acids is clearly more intense than in the
spectrum of primary organic aerosols; c) the pattern of bands at 0.9, 1.3, 1.6 and 2.2-2.4 ppm highlights the
presence of linear aliphatic structures substituted with oxo- and carboxylic groups. Nevertheless, MSA and
the low molecular weight amines were less abundant in A-2401 than in the sample from the Weddell Sea
(Table 1). Also the ratio between acyl (C$\underline{H}$-C=O) and alkyl (C$\underline{H}$-CH) groups was smaller in A-2401 than in A-





0911 (Figure S8). The linear aliphatic structures involved longer methylenic chains in A-2401 than in A-
0911, so that in the former case they were more similar to the aliphatic structures of the aerosolized melted sea
ice (Figure S8). Another difference between the two ambient aerosol samples is that the one from the
Southern Ocean contains much more alcoxy groups (HC-O, in the chemical shift range 3.4 – 4.2 ppm) of
polyols than the one from the Weddell Sea (Figure 5; Table 1). When comparing the functional group
distributions of the ambient aerosol samples to that of the aerosol generated during the tank experiments,
clearly the samples from the Southern Ocean show a better match than the samples from the Weddell Sea
do. Other similarities between the composition of A-2401 and the aerosol in the tank can be found in the
fine structures of the spectra, especially in the ranges of aromatics, acetals and polyols (Figure S12). A-2401
clearly contains traces of organic markers of primary aerosols and specifically glycerol, N-osmolytes (Figure
S10) and aminoacids (alanine). Finally, contrary to A-0901, sample A-2401 contains only trace amounts of
creatinine.

**3.3.3. Ambient aerosols influenced by coastal land sources.**
Sample A-0701 was collected in the western sector of the Weddell Sea. The air masses showed several
overpasses on the Antarctic Peninsula. The [1]H-NMR spectrum shows unique features: isobutyric acid was
found in relatively high concentrations, together with an amine tentatively identified as cadaverine (Figure
5). The aliphatic chains occur in much lower amounts than in the samples described above, the band of
acyls is not as pronounced as in A-0901 (Figure S8), whereas alcoxyls are abundant, especially due to the
occurrence of sucrose at a remarkable concentration of 10 ng/m$^3$. Finally, no creatinine was found in this
sample. Clearly, the composition of A-0701 is drastically different from that of the other samples collected
in the Weddell Sea. The presence of sucrose (Figure S9) points to a contribution from primary biological
particles emitted from a terrestrial biota, not a marine one. Vegetation cover (scarce but existing) in the
Antarctic Peninsula can be responsible for such emissions. The NMR composition of A-0701 provides
evidence of the diversity of biogenic aerosol sources active in this area of the world.

**4. Discussion**
**4.1  Source apportioning of primary and secondary organic components in different regions**
The comparison of the NMR compositions of the ambient aerosol samples collected onboard *RV Hesperides*
(Figure 8) supports the distinction of aerosol sources between the sympagic and pelagic environments
already introduced by Dall'Osto et al. (2017). The higher abundance of alkyl (C-H) and alcoxy (H-C-O) groups
detected in the second half of the cruise points to a larger fraction of primary organic compounds rich in
lipids and polyols in the aerosols of the open Southern Ocean. Analogous compositions were obtained
using FTIR spectroscopy at Ross Island (Liu et al. 2018). In our study, the attribution of compound classes
and molecular markers (such as glycerol and N-osmolytes) to primary marine particles was supported by
the comparison with the analysis of tank-generated sea-spray particles. According to our NMR datasets,
primary marine organics were ubiquitous in the region as witnessed by the presence of glycerol in all
samples. However, glycerol accounted for almost the entire polyol content in the three samples from the
eastern/north Weddell Sea, while the samples from the open ocean contained much larger and more
complex mixtures of polyols/sugars. Sub-ng/m$^3$ levels of free aminoacids (alanine) and trace amounts of N-
osmolytes along with the greater abundance of linear aliphatic structures similar to lipids in the samples
from the Southern Ocean point to a major contribution of primary organics to submicron organic aerosols
in this environment. These findings provide further confirmation to the importance of sea spray as a source
of marine organic particles in oceanic regions characterized by high productivity and strong wind stress.





In sympagic waters, other mechanisms of aerosol formation take place. Sympagic waters are rich in S- and
N- osmolytes produced by the algal communities colonizing the sea ice. The osmolytes degrade to VOCs
which are then converted to SOA components, such as MSA (Davison et al., 1996) and low-molecular
weight methyl-amines (Facchini et al., 2008). Also the distribution of the oxygenated functional groups was
different between sympagic and pelagic regimes. If alcoxyl groups (H-C-O) from polyols and sugars
accounted for almost 50% of total alcoxyl (H-C-O) and acyls (H-C=O) in the samples from the Southern
Ocean, such fraction decreased to less than 30% in the three samples from the offshore areas of the
Weddell Sea (Fig. 8). The mixtures of organic compounds carrying acyls, like carboxylic and oxocarboxylic
acids, are not associated to primary marine aerosols and are likely components of SOA. Carboxylic acids can
form photochemically (Cui et al., 2019) during the austral summer. The nature of parent VOCs for
carboxylic acids in our samples is unknown, but the occurrence of linear aliphatic compounds containing
oxo- and carboxylic groups indicates that one of the possible sources stands in the oxidative degradation of
lipids - either in the aerosol or in the marine microlayer - as suggested by past studies in Antarctica
(Kawamura et al., 1996) and consistent with recent AMS observations in the Arctic marginal ice zone (Willis
et al., 2017).
In the Weddell Sea, under the influence of air masses that had travelled over the Peninsula (sample A-
0701), the contribution of the emissions from the land biota became evident, therefore supporting the
observations of Schmale et al. (2013) on the contribution of primary biological particles from the coastal
land ecosystems. Our data suggest that beside animal colonies, also the land vegetation (grasses, mosses,
lichens) of the Antarctic Peninsula can contribute to the emission of particles, and in particular to the
content of sugars. Other biological compounds of primary origin, the aminoacids, were not found in the
Weddell Sea in our study. These results contrast with the previous findings that a significant fraction of the
ambient PM1 mass was accounted for by proteinaceous material at an island site in the Southern Ocean
(Schmale et al., 2013). On the other hand, the observations of Schmale et al. (2013) were carried out under
the direct influence of the emissions of seabird colonies, while our observations were carried out offshore.
More research is needed to quantify the range and extent to which primary particles from the terrestrial
biota impact the marine aerosol composition in the Antarctic region.
**4.2 A new potential marker: creatinine.**
The sources of creatinine in the ambient aerosol is controversial. On the basis of its chemical structure, it is
water-soluble but clearly less volatile than the methyl-amines and, as a consequence, its Henry coefficient
must be much less favorable for transferring this amine out of seawater into the gas phase. A primary origin
via sea spray is also doubtful because creatinine is not a strong surfactant. On the other hand, Prather et al.
(2013) showed that sea-spray aerosols encompass several classes of organic particles, including some made
of biological material: POC particles and large colloids can be scavenged by rising bubbles and injected in
the atmosphere by jet drops. Jet drop emission represents a plausible mechanism to transfer primary
organic compounds which are not strong surfactants from seawater to the atmosphere. If this happened to
creatinine, it must have occurred in source areas other than the algal blooms where we conducted the tank
experiments, since we did not detect any creatinine in the aerosolized sea water and sea ice. Creatinine is a
common metabolite of mammals, therefore an alternative source via the excreta of sea lions in Antarctic
coastal areas can be postulated. However, a much more vast potential source in seawater is also possible
under the hypothesis that creatinine results from the enzymatic conversion of creatine, which is a known
metabolite of the urea cycle in marine animals (Whitledge and Dugdale, 1972) and phytoplankton (Allen et
al., 2011) that contributes to pelagic DOC across the world's oceans (e.g., Wawrik et al., 2017).
**5. Conclusions**



Our results demonstrate that, beside MSA, a complex mixture of biogenic organic compounds contributes
to the composition of submicron aerosol particles in the Antarctic atmosphere. Although individual organic
markers encompassing sugars, aminoacids and carboxylic acids have already been identified in past studies,
our results indicate that non-MSA biogenic organic compounds impact the bulk composition of organic
aerosol in this environment (Figure 8). The NMR analysis provides evidence for both secondary (more
important in sympagic regions) and primary marine (more important in pelagic areas) sources. A third
contribution from the terrestrial biota in the Antarctic Peninsula was also identified. The emission of sea-
spray organics in offshore areas was unambiguously demonstrated by the determination of molecular
tracers for lipids and polyols and by the comparison of the fine structures in the $^1$H-NMR spectra of the
ambient samples and of the aerosol generated in the tank experiments. A new biogenic marker, creatinine,
was identified for the first time in the ambient aerosol, extending the list of reduced nitrogen containing
molecular tracers in the atmosphere. The discovery of creatinine also exemplifies the usefulness of
employing non-targeted analytical techniques like NMR spectroscopy for screening the organic composition
of the aerosol in remote environments where the sources of atmospheric particulate matter are still poorly
known. The complexity of the organic composition illustrated in this study calls for more research on
suitable methodologies – both online and offline and combinations of them – to investigate the nature of
non-MSA marine organic particles in off-shore regions around the Antarctic continent.

**Data Availability**
The NMR data sets are available on request to the corresponding author.

**Author Contribution**
SD wrote the paper; MDO and RS coordinated the experimental activities in the field; MP, MDO and SG
collected the aerosol samples; MDO, MP and DC collected the sea ice samples; COD, JO and DC set up the
bubble bursting tank; MR, MP, MR, NZ and FV performed the sample extraction and preparation for WSOC
and NMR analysis; NZ and MP performed the NMR analyses; SG and CJK carried out the HPLC/MS analyses;
SD, MP and ET elaborated the NMR data; MDO, RS and ET contributed to the interpretation of the analyses
of the seawater samples; SD, MP, MR, MDO, TH, CJK and ET contributed to the interpretation of the
analyses of the aerosol samples; all authors contributed to the general discussion and to work out the main
conclusions of this study.

**Acknowledgments**
The cruise was funded by the Spanish Ministry of Economy through projects PEGASO (CTM2012-37615) and
Bio-Nuc (CGL2013-49020-R). The research leading to these results has received funding from the European
Union's Seventh Framework Programme (FP7/2007-2013) Project BACCHUS under Grant Agreement
603445. The research activities of CNR were also supported by the project AirSEaLab: Progetto Laboratori
Congiunti. We would like to thank Prof. Andrea Mazzanti for his advice in performing the NMR experiments
at the NMR facility of the Dep. Industrial Chemistry, University of Bologna. We also thank Dr. David
Beddows (Uni. Birmingham) for help in drawing figures, in particular air mass back trajectories.

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





**Table 1.**

| sample ID: | **A-0701** | **A-0901** | **A-1301** | **A-1801** | **A-2401** | **A-2801** | **A-0102** | **A-0602** |
|---|---|---|---|---|---|---|---|---|
| sampling times: | 07 Jan 20:00 – 09 Jan 09:00 | 09 Jan 14:50 – 13 jan 13:50 | 13 Jan 19:20 – 18 Jan 12:20 | 18 Jan 13:30 – 21 jan 23:55 | 24 Jan 15:00 – 28 jan 05:15 | 28 Jan 13:30 – 31 jan 13:50 | 01 Feb 14:50 – 06 Feb 03:15 | 06 Feb 22:00 – 10 Feb 11:00 |
| average air mass type: | Weddell Sea / Antarctic Peninsula | Weddell Sea | Weddell Sea | Weddell Sea | Open Ocean | Open Ocean | Open Ocean / mixed | Open Ocean |
| *Water-soluble organic carbon (µgC/m³):* | | | | | | | | |
| WSOC | 0.14 | 0.07 | 0.12 | 0.13 | 0.09 | 0.14 | 0.05 | 0.11 |
| *¹H-NMR functional groups (nmolH/m³):* | | | | | | | | |
| H-C | 2.60 | 2.16 | 2.28 | 3.03 | 3.27 | 2.81 | 2.07 | 2.82 |
| H-C-C=O | 2.40 | 1.58 | 1.80 | 2.10 | 1.91 | 1.86 | 1.28 | 1.78 |
| H-C-O | 2.15 | 0.57 | 0.69 | 0.83 | 2.06 | 0.99 | 0.99 | 1.41 |
| O-CH-O | 0.20 | 0.07 | 0.05 | 0.04 | 0.08 | 0.09 | 0.07 | 0.09 |
| Ar-H | 0.12 | 0.05 | 0.00 | 0.10 | 0.09 | 0.10 | 0.11 | 0.07 |
| MSA | 2.13 | 1.95 | 2.63 | 4.54 | 1.72 | 2.90 | 2.22 | 1.53 |
| Alkyl-Amines | 0.30 | 0.79 | 0.53 | 1.32 | 0.34 | 0.49 | 0.13 | 0.15 |
| *Molecular markers (ng/m³):* | | | | | | | | |
| MSA | 68 | 62 | 84 | 145 | 55 | 93 | 71 | 49 |
| methyl-amines | 2.31 | 5.5 | 3.79 | 9.0 | 2.53 | 3.56 | 0.92 | 1.20 |
| creati-nine | 0.09 | 1.65 | 1.52 | 2.21 | ~0.05 | 1.00 | 0.29 | 0.41 |
| glycerol | NA | 1.1 | 0.7 | 0.7 | 3.0 | 0.8 | 0.7 | 1.3 |
| sucrose | 11 | | | | | | | |
| alanine | | | traces[1] | traces[1] | 0.6 | | | 0.7 |
| betaine | | | | | traces[2] | | | |

[1] below the limit of quantification (0.3 ng/m³); [2] below the limit of quantification (0.2 ng/m³)

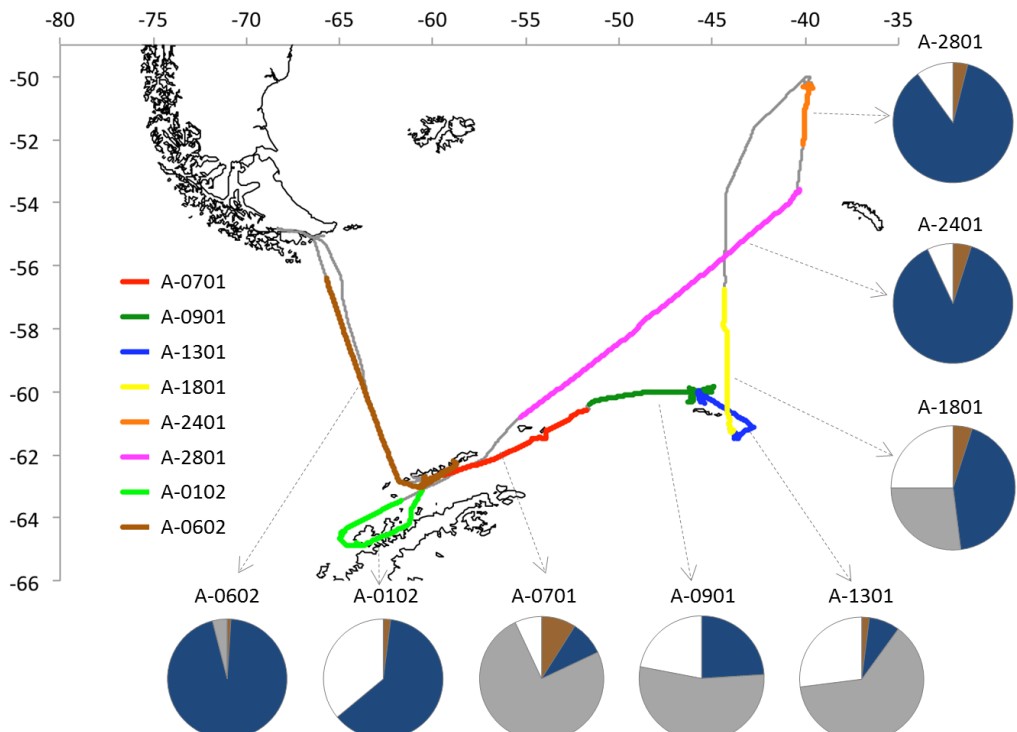

**Figure 1**. Cruise of RV Hesperides. The colors indicate the duration of the single aerosol samplings (short interruptions undertaken to avoid contamination from ship emissions are not indicated in the figure). The average time spent by air masses travelling over land (brown), marginal ice zone (1-99% surface coverage; grey), compact sea ice (100% coverage; white) and open ocean (dark blue) is indicated for each sample.





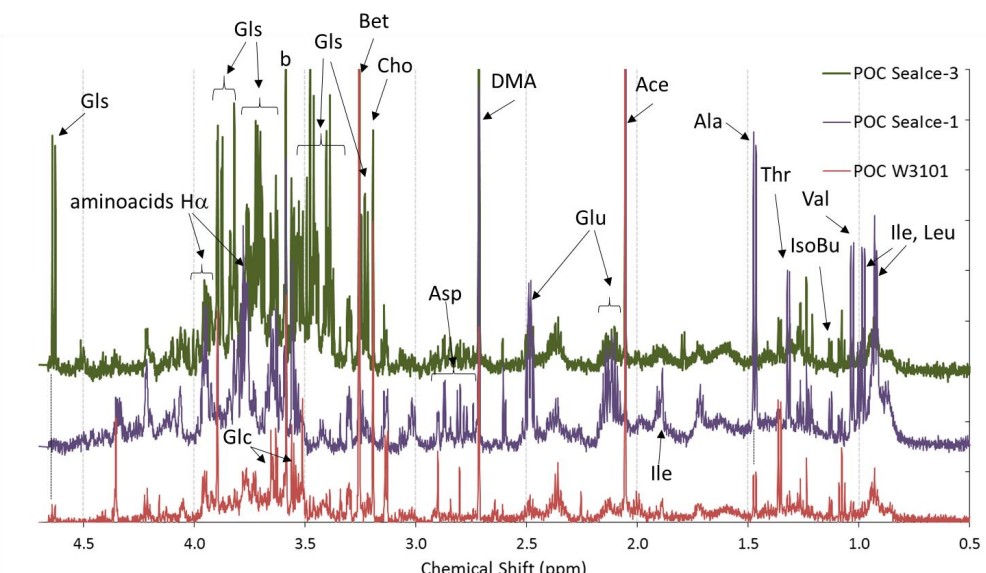

**Figure 2**. Aliphatic region of the ¹H-NMR spectra of three POC sample extracts: one for the seawater sample (POC W3201) and two from melted sea ice (POC SeaIce-1 and POC SeaIce-3). Specific NMR resonances were assigned to: the residuals of aminoacids (Ala, Thr, Val, Ile, Leu, Glu and Asp) and their alpha hydrogen atoms, isobutyric acid (IsoBu), acetic acid (Ace), dimethylamine (DMA), N-osmolytes (Bet: betaine; Cho: choline), glycerol (Glc) and to glucose (Gls).

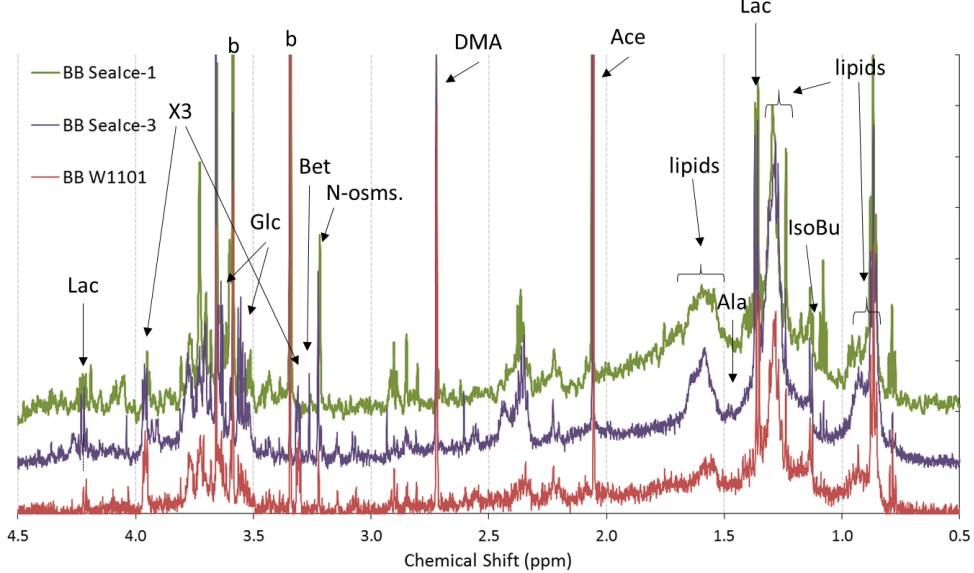

**Figure 3**. The same as Figure 2 but for the three bubble bursting aerosols: from seawater sample W1101 (BB W1101) and melted sea ice #1 and #3 (BB SeaIce-1 and BB SeaIce-3). Specific resonances were assigned to lactic acid (Lac), acetic acid (Ace), isobutyric acid (IsoBu), alanine (Ala), dimethylamine (DMA), glycerol (Glc), N-osmolytes (Bet: betaine;




"N-osms": unidentified, possibly phosphocholine) and to blank contaminations (b). Unresolved mixtures of aliphatic compounds were identified as lipids.

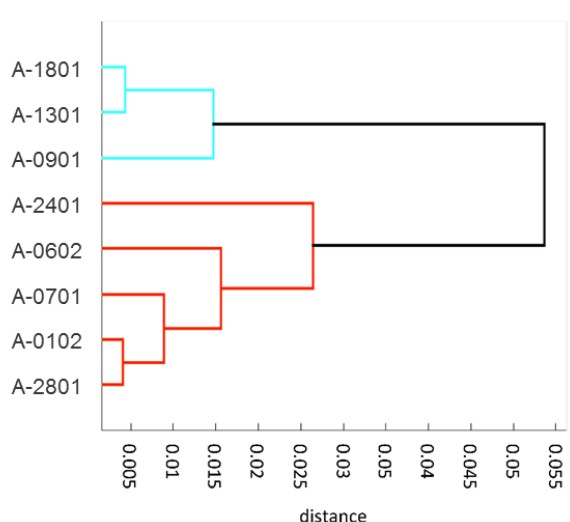

**Figure 4**. Cluster analysis of the ¹H-NMR spectra of the PM1 HiVol samples of ambient aerosol.

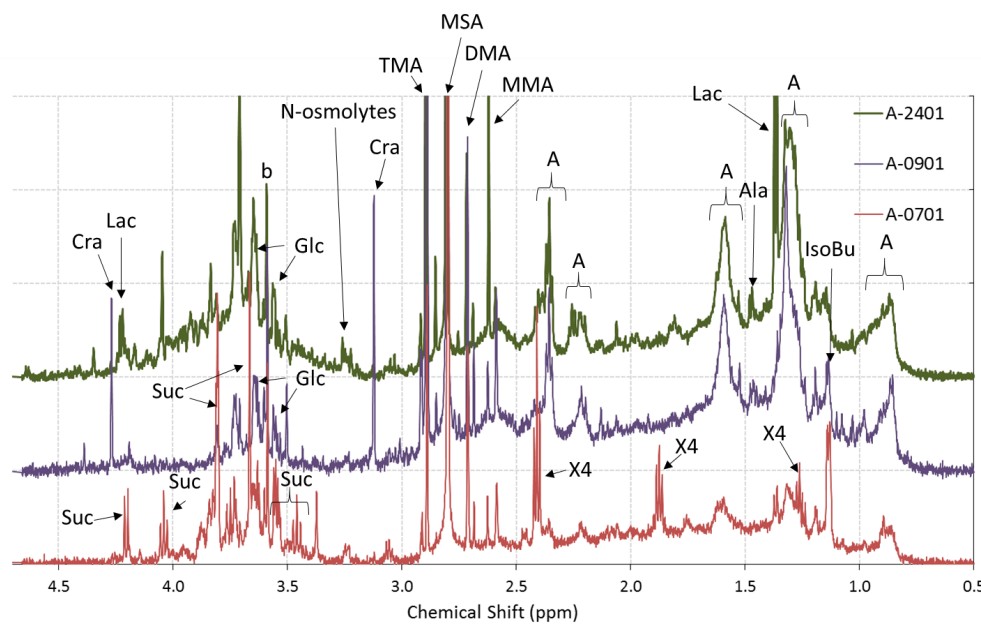

**Figure 5**. The same as Figure 2 but for the three ambient submicrometer aerosol samples. Specific resonances were assigned to lactic acid (Lac), isobutyric acid (IsoBu), alanine (Ala), monomethylamine (MMA), dimethylamine (DMA), trimethylamine (TMA), glycerol (Glc), sucrose (Suc), creatinine (Cra) and to blank contaminations (b). Unresolved



mixtures of linear aliphatic compounds (A), including possible contributions from lipids, are indicated in the spectra. Other NMR signals were only tentatively attributed to cadaverine (X4).

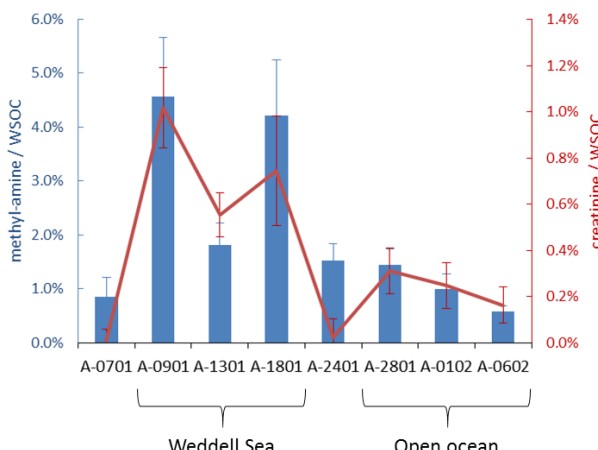

**Figure 6**. Concentrations of creatinine and methylamines in the PM1 samples. The concentrations are expressed as contributions to WSOC (mol% of carbon). "Weddell Sea" and "Open ocean" labels indicate the sampling periods identified by Dall'Osto et al. (2017) to characterize the aerosol composition in air masses travelling over sea ice and in the Southern Ocean, respectively.



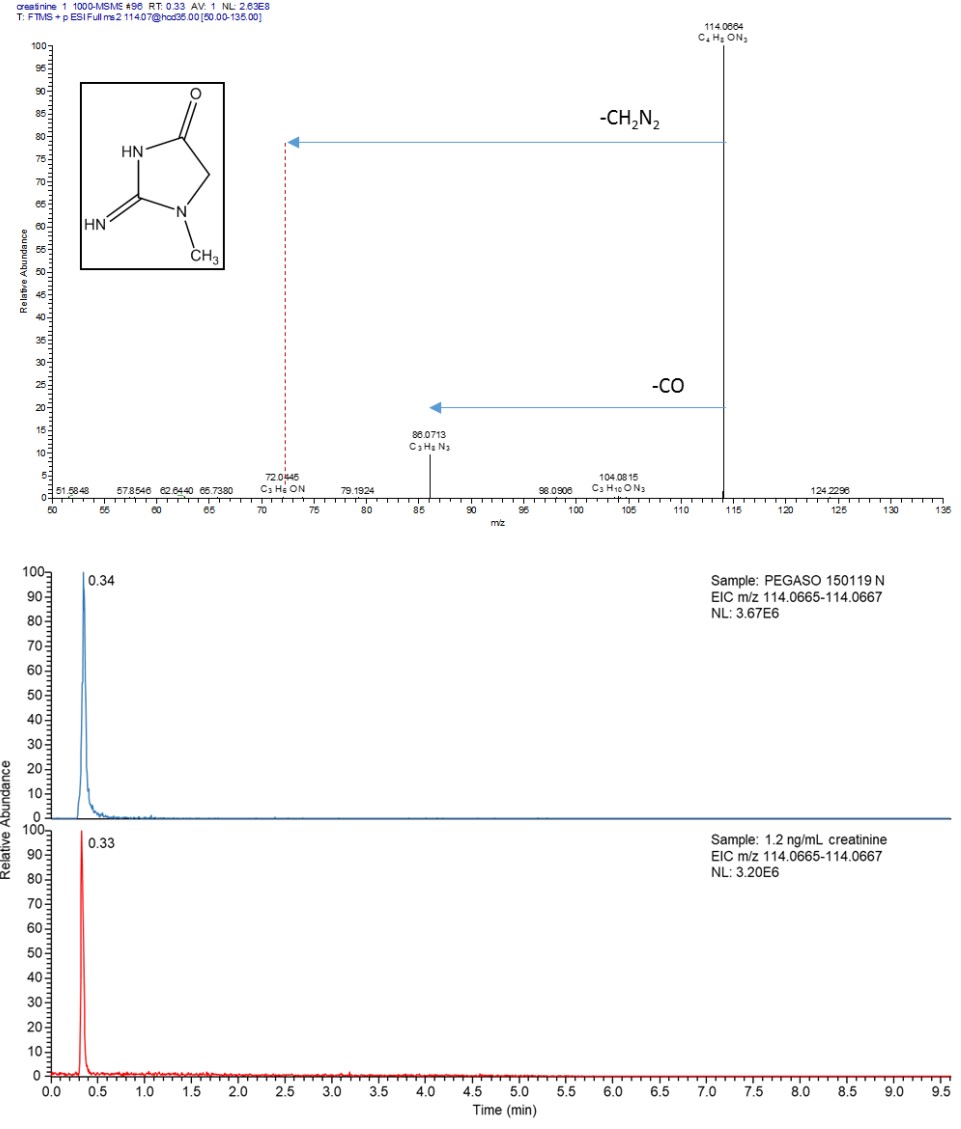

**Figure 7**. (top) MS spectrum of a creatinine standard. (bottom) Extracted ion chromatograms for m/z 114.0655-114.0667, corresponding to creatinine, of the filter extract of sample 0119n obtained during the PEGASO campaign and the neat creatinine standard. The retention time of creatinine was found to be 0.33 min using the conditions outlined in Section 2.4.





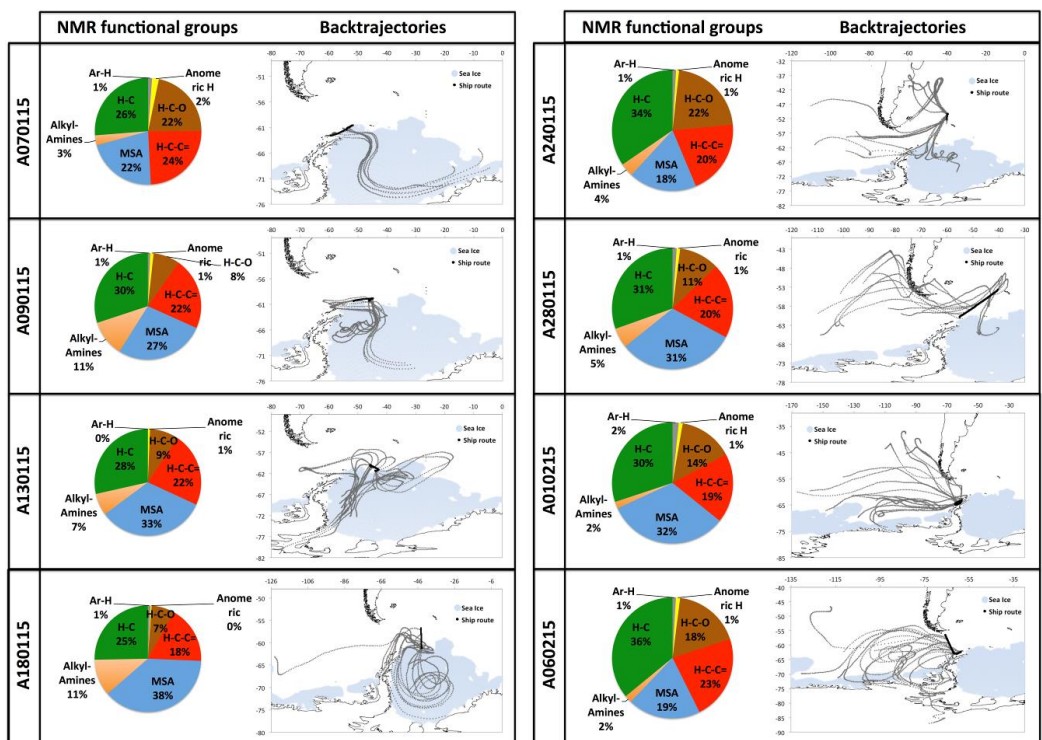

**Figure 8**. NMR functional group compositions of WSOC in the PM1 HiVol samples. Functionalities: H-C (alkyls), H-C-(C=) (acyls), H-C-O (alcoxyl), MSA, amines, anomeric, Ar-H (aromatic).