# Peer review of "Shipborne measurements of Antarctic submicron organic aerosols: an NMR perspective linking multiple sources and bioregions"

_Atmospheric Chemistry and Physics, 2019_

## Referee Comment (RC1) · Anonymous Referee #1 · 18 Dec 2019

Review of 'Shipborne measurements of Antarctic submicron organic aerosols: an NMR perspective linking multiple sources and bioregions' by S. Decesari et al.

The paper by S. Descesari et al. deals with the composition of marine aerosols in ambiant antarctic air and artificially generated from ambiant seawater as nascent sea spray, in relation with seawater composition. In particular the organic fraction of marine aerosol is analyzed with a high precision method, providing unique information on the presence of lipids, sugars and proteins in marine organic aerosols. Conclusions can be drawn on the contribution from primary vs secondary sources in ambiant antarctical aerosol. Understanding the origin of the marine organic aerosol is of importance for

climate models, as the prediction of CCN and IN of biological origin are mostly based this fraction emitted to the atmosphere. Although the introduction is very well documented, it could be completed with a few lines on why organic marine aerosols are important. Also, since the work presented here follows the publication by Dal' Osto 2019, it would be useful to summarize the Dal'Osto main conclusions and what the present work adds to them already in the introduction. Despite these remarks, because the data presented originate from a poorly characterized region, I recommend that the paper is published after minor revisions that I will detail below.

Major comments

1-Are the nascent sea spray generation experiments running with the same seawater (closed loop) or continuously flushed with fresh seawater? If performed in a closed loop fashion, what impact on an eventual depletion of surfactant organics from the sample with time? I it said that 9 samples were performed but only one sample is analyzed by HNMR. It would be useful to state what was the spatial variability of the general chemical composition of these 9 samples, and how the one sample analyzed with HNRM compared to the rest of the samples.

2-In general, variability among samples is not discussed much neither for the seawater samples. What differences amongst the 45 POC samples of seawater? Is a comparison between bloom versus non bloom POC content possible?

3-For comparing aerosol Organic carbon characteristics with those of organic carbon in the seawater, the results on the seawater DOC analysis should be known as both are expected to contribute to the aerosol organic matter. As these analysis are presumably not available, there should be a discussion on the fact that the POC 10-45micron composition does not represent the full organic mater present in the seawater. This has implications on the conclusions made on preferential organic transfer to the atmosphere.

4- Again, there is only one sample or primary marine aerosol (PMA) generated from

the tank experiment, so we do not know the variability of the organic composition of PMA in this region. This should be discussed, especially when stating that creatinine was not detected in the PMA (this could be true for the one sample presented but not for the others..?) .

Minor comments

Line 3: two ways Line 30: century Line 134: already specified line 133 Line 255 : do you mean DMSP (and not DMPS) ?

---

## Referee Comment (RC2) · Anonymous Referee #2 · 22 Dec 2019

This manuscript provide the detail information about ambient aerosol composition and possible origin of the aerosol in Antarctic region throughout state-of-the-art high resolution NMR and HRHPLC-MS analysis with the composition of marine aerosols and bubbling aerosol generated from suspended seawater. This paper especially provide unique information on the presence of creatinine as a new marker of marine organic aerosols. Understanding the origin of the marine organic aerosol is of importance to clearly understand the effect of aerosol on climate change, especially in Antarctic and Artic region. However, the limitation of assessment of these region, the studies relating to the understanding physical and chemical characteristics of aerosol in these regions are poorly understood. Thus, I believe this manuscript provide the valuable data and

result to solve the puzzle for organic aerosol composition in Antarctic region. So, I think this paper is suitable to be published in ACP with minor revision that I will comment in below.

Comment and Suggestion. 1. In section 2.5 of the manuscript, the details for the operating method and condition of UHPLC-HESI-Orbitrap-MS are provided. However, the information for the target compounds including creatinine is not provided. The analytical method for the target compounds including calibration results, QA/QC should be included in this section. 2. This study analyzed seawater generating aerosol using bubble chamber. However, the methodology and information how to generate aerosol from seawater is not available in this manuscript. 3. In 3.3.1. Ambient aerosols from the Weddell Sea: Check the sample labeling. In Figure 1. There are no information for sample A-0911 4. English expression is ambiguous in the manuscript. Please revise the whole of the paper to improve English expression.

---

## Author Comment (AC1) · 4 Feb 2020

**REPLY to Anonymous Reviewer #1**

We thank the Reviewer for the positive comments. We accept her/his invitation to integrate the Introduction highlighting the importance of organic marine aerosols along with a summary of Dall'Osto et al. 2017 reporting the first results of atmospheric and seawater measurements carried out during the PEGASO cruise.

The following text was added to the manuscript:

(After line 50 in the Introduction): "DMS and other reactive volatile species are known precursors to the secondary marine aerosol which contribute to the aerosol population in the marine boundary layer together with primary sea-spray particles. Marine aerosols impact global climate by reducing the amount of solar radiation reaching dark surface of the ocean, both directly (through scattering) and indirectly (by modulating cloud formation and life-time) (O'Dowd and de Leeuw 2007). Furthermore, in polar regions, cloud seeding by marine aerosols transported over glaciated regions also affects the longwave radiation budget (Willis et al. 2018)."

And lines 58 to 64 of the Introduction are rewritten with a more detailed summary of the Dall'Osto et al. (2017) study including information about the tank experiments which are key in this study:

"By contrast, other biological parameters of seawater, like chlorophyll a, total organic carbon (TOC) and transparent exopolymeric particles (TEP), showed higher concentrations in the open Southern Ocean north of the SBACC. Results of bubble bursting experiments conducted on nascent seawater as well as using melted sea ice showed that organic nitrogen and organic carbon were more abundant in the aerosol in the latter case. Moreover, the production of organic-rich particles was better traced by markers of the ice biota, such as mycosporines, than by macro-tracers of biological productivity (chlorophyll). These results indicate that not only productivity per se but also the composition and ecophysiological state of the microbiota affect the production of aerosol precursors in seawater. Indeed, the observations of organic nitrogen in the aerosol – carried out by both online and offline chemical methods – pointed to strong sources in the area of the Weddell Sea which, at the time of the field campaign, was heavily covered by sea ice."

Replies to the specific Referee's comments are provided below:

*1-Are the nascent sea spray generation experiments running with the same seawater (closed loop) or continuously flushed with fresh seawater? If performed in a closed loop fashion, what impact on an eventual depletion of surfactant organics from the sample with time? I it said that 9 samples were performed but only one sample is analyzed by HNMR. It would be useful to state what was the spatial variability of the general chemical composition of these 9 samples, and how the one sample analysed with HNRM compared to the rest of the samples.*

**REPLY:** The bubble bursting experiments with seawater were conducted in the tank continuously flushed with a fresh sea water provided by the ship's underway pumping system resulting in the water residence time inside the tank of approximately 10-20min. The sea-ice experiments were instead performed in the tank in a closed loop fashion because of the limited amount of water volume available from the melted sea ice samples. We agree with the Referee that under such conditions, modifications in seawater can be induced by the forced aerosolization itself, with dependence on the technical characteristics of the apparatus for bubble bursting. During past sea-spray generation experiments with the same equipment and closed loop system, the online monitoring of organic aerosol concentrations did not show any decline with

time (O'Dowd et al., 2015) with no evidence of surfactant depletion effects of the film. Therefore, it is unlikely that the depletion could have occurred in a continuously flushed system. However, the focus of the present study is not organic enrichment factor of sea spray aerosol but rather its organic composition. We will insert a new short paragraph to comment the possible effects of the bubble bursting experimental conditions:

(appending after line 134 of Section 2.2) "In six cases, bubble bursting experiments were conducted in the tank continuously flushed with fresh seawater conveyed form the ship's pumping system. In the three sea-ice experiments instead bubble bursting was carried out in a closed loop system because of the limited amount of water volume available from the melted sea ice samples. In this case, the bubble bursting process could lead to chemical and biological modifications in the samples like a progressive depletion of surfactants on the film. Quantification of such artefacts is unavailable. Nevertheless, past studies carried out in different geographical region of North East Atlantic but with the same apparatus showed no evidence of decreasing organic enrichment in the generated sea spray when operated in a closed loop system (O'Dowd et al., 2015)."

The nine bubble bursting experiments conducted during the campaign included 6 carried out with seawater (in a continuously flushed tank) and 3 with melted sea ice (in a closed loop tank). The three sea ice samples are the same discussed in Dall'Osto et al. (2017). Two of the three experiments provided enough material for off-line chemical analysis by $^1$H-NMR spectroscopy, and these are the samples discussed in the present manuscript (BB SeaIce-1 and BB SeaIce-3). All the samples of sea ice were collected at the northern edge of the Weddell Sea marginal ice zone, south of South Orkney islands. The other six samples from the tank were obtained by bubble bursting of seawater and a summary of the online aerosol measurements is also included in Dall'Osto et al. (2017). Water was collected mostly in highly productive oceanic regions from diverse geographical areas: from the blooms by the South Orkney, to those nearby South Georgia, and finally in the highly productive coastal areas of the Antarctic peninsula. Only during one of the bubble bursting experiment, more oligotrophic waters were collected during the transect from the South Georgia to the Antarctic Peninsula. The sample selected for NMR analysis (BB W1101) was obtained from seawater in a bloom area west and north of the South Orkney. We will include a short explanation in the new version of the paper:

(At line 231 at the beginning of Section 3.2): "The three primary marine aerosol samples collected in the tank and analysed by $^1$H-NMR spectroscopy included the following samples. One sample was collected from bubble bursting of nascent seawater (BB W1101) obtained during almost four days of navigation west and north of the South Orkney Islands with seawater continuously flushed onboard the RV maintaining continuous sea spray production in the tank. The other two samples (BB SeaIce-1 and BB SeaIce-3) were obtained from two of the three sea ice samples melted in the tank and run in a closed loop system. Sea ice was collected from the marginal ice zone around 100 km south of the South Orkneys by using small inflatable boats and clean laboratory ware. The chemical information obtained for these bubble bursting aerosols is, therefore, representative for primary marine particles in the northern sector of the Weddell Sea."

*2-In general, variability among samples is not discussed much neither for the seawater samples. What differences amongst the 45 POC samples of seawater? Is a comparison between bloom versus non bloom POC content possible?*

REPLY: The four samples analysed by $^1$H-NMR spectroscopy had a similar POC content: 10, 12, 9.8 and 11 μmolC L$^{-1}$. This corresponds to ca. to the 75%-percentile of the POC distribution of the 45 POC samples (average ± standard deviation for the full set was 8.7 ± 4.1 μmolC L$^{-1}$). All four selected samples originated from bloom areas, where POC concentration ranged between 8 and 12 μmolC L$^{-1}$ (with peaks above 15

µmolC L$^{-1}$), while the oligotrophic areas of the Southern Ocean showed concentrations of about 4 µmolC L$^{-1}$. The text in section 3.1 (lines 203 to 207) was modified and integrated as follows:

"Figure 2 shows the proton NMR spectra of three POC samples, one from seawater (POC W3101) and two from melted sea ice (POC SeaIce-1, and POC SeaIce-3) as examples. It is worth noting that the samples were pre-filtered through a polycarbonate membrane of 10 µm porosity, hence the analyzed POC fraction represents only the fine fraction (between ~ 0.45 and 10 µm). During PEGASO, the concentration of fine POC fraction (0.45 – 10 µm) ranged between 8 and 12 µmolC L$^{-1}$ in bloom areas. The sub-set of samples analysed by $^1$H-NMR spectroscopy exhibited a concentration of 10.6 ± 0.7 µmolC L$^{-1}$ (n = 4). Sample POC W3101 originated from the bloom area west of South Georgia island, while the two sea ice samples were collected in the marginal ice zone of the Weddell Sea. The interpretation of the spectra was carried out by comparison with the datasets and spectra provided in the literature on metabolomics…."

*3-For comparing aerosol Organic carbon characteristics with those of organic carbon in the seawater, the results on the seawater DOC analysis should be known as both are expected to contribute to the aerosol organic matter. As these analysis are presumably not available, there should be a discussion on the fact that the POC 10-45micron composition does not represent the full organic matter present in the seawater. This has implications on the conclusions made on preferential organic transfer to the atmosphere.*

**REPLY**: The concentrations of POC$_{0.45-10µm}$ determined in this study (8.7 ± 4.1 µmolC L$^{-1}$) represents a small amount with respect to the TOC concentrations reported in Dall'Osto et al. (2017) (50 – 70 µmolC L$^{-1}$), and we agree with the Reviewer that a large fraction of TOC could not be characterized by our analytical techniques. Nevertheless, not all TOC components can contribute to sea spray composition. The pre-filtration of our seawater samples through 10 µm -pore membranes was carried out with the purpose of excluding POC particles that are too large to form primary marine aerosols. The DOC pool as well includes compounds, like marine refractory fulvic material, which are homogenously distributed in the water column with a small enrichments in the surface film (Hertkorn et al., 2013). We will specify the limitations of our methodology with respect to seawater sample analysis as follows:

(To append after line 202 in Section 3.1): "The chemical characterization of the smallest POC component (0.45 – 10 µm) aims to provide information about composition of the buoyant particles, while the contribution from DOC to the surface film composition could not be determined in this study."

*4- Again, there is only one sample or primary marine aerosol (PMA) generated the tank experiment, so we do not know the variability of the organic composition of PMA in this region. This should be discussed, especially when stating that creatinine was not detected in the PMA (this could be true for the one sample presented but not for the others..?).*

**REPLY**: We agree with the reviewer. Nevertheless, the single PMA sample derived from bubble bursting in seawater shows very similar spectroscopic features with the other two PMA samples obtained by aerosolizing melted sea ice (Figure 3). Sample BB W1101 was obtained from sea water collected in an area west and north of the South Orkney islands, approximately 100 km north of the edge of the marginal sea ice zone where the ice samples were subsequently collected. Our data, though based on a small sample number (but sample BB W1101 corresponds to almost four days of navigation), indicate that sea spray aerosol in the northern sector of the Weddell Sea does not contain creatinine. However, little can be said about the PMA composition from other geographical regions of the Southern Ocean (South Georgia, etc.). This will be acknowledged in the revised version of the manuscript. We copy the proposed text from above in relation to Major comment #1:

(At line 231 at the beginning of Section 3.2): "The three primary marine aerosol samples collected in the tank and analysed by [1]H-NMR spectroscopy included the following samples. One sample was collected from bubble bursting of nascent seawater (BB W1101) obtained during almost four days of navigation west and north of the South Orkney Islands with seawater continuously flushed onboard the RV maintaining continuous sea spray production in the tank. The other two samples (BB SeaIce-1 and BB SeaIce-3) were obtained from two of the three sea ice samples melted in the tank and run in a closed loop system. Sea ice was collected from the marginal ice zone around 100 km south of the South Orkneys by using small inflatable boats and clean laboratory ware. The chemical information obtained for these bubble bursting aerosols is, therefore, representative for primary marine particles in the northern sector of the Weddell Sea."

All minor Reviewer comments have been addressed accordingly.

**References:**

Dall'Osto et al., Antarctic sea ice region as a source of biogenic organic nitrogen in aerosols, Scientific Reports, 7, 6047, 2017.

Hertkorn et al., High-field NMR spectroscopy and FTICR mass spectrometry: powerful discovery tools for the molecular level characterization of marine dissolved organic matter, Biogeosciences, 10 (3), 1583–1624, 2013.

O'Dowd, C. D. & de Leeuw, G. Marine aerosol production: a review of the current knowledge. Philosophical Transactions of the Royal Society A: Mathematical, Physical and Engineering Sciences 365, 1753, 2007.

O'Dowd et al., Connecting marine productivity to sea-spray via nanoscale biological processes: Phytoplankton Dance or Death Disco? Scientific Reports, 5, 14883, 2015.

Willis et al., Processes controlling the composition and abundance of Arctic aerosol. Reviews of Geophysics, 56, 621–671. https://doi.org/10.1029/2018RG000602, 2018.

---

## Author Comment (AC2) · 4 Feb 2020

**REPLY to Anonymous Reviewer #2**

We thank the Reviewer for the very positive comments. We report our point-by-point response to the specific comments raised in her/his review:

*1. In section 2.5 of the manuscript, the details for the operating method and condition of UHPLC-HESI-Orbitrap-MS are provided. However, the information for the target compounds including creatinine is not provided. The analytical method for the target compounds including calibration results, QA/QC should be included in this section.*

Calibration results of the UHPLC-HESI-HRMS measurements are now given in a more detailed manner in the Supplement, including calibration function, regression coefficient and retention times (new table S2):

"Table S2. Creatinine calibration results by UHPLC-HESI-HRMS

| Calibration std. | Conc. (ng/mL) | Peak area (a.u.) | RT (min) |
|---|---|---|---|
| 1 | 1.2 | 6302965 | 0.33 |
| 2 | 12 | 56953602 | 0.33 |
| 3 | 120 | 440017098 | 0.33 |
| Calib. Function* | (3.61E6±0.009E6)x | +(7.53E6±6.21E6) | |
| R² | 0.999 | | |

*Linear least squares fit in MS Excel 2010"

*2. This study analyzed seawater generating aerosol using bubble chamber. However, the methodology and information how to generate aerosol from seawater is not available in this manuscript.*

The methodology is described in the second paragraph of Section 2.2. We included a more comprehensive description clarifying the protocols used to produce the samples discussed in this study in comparison with the methodology employed for the online measurements discussed in the previous publication by Dall'Osto et al. (2017):

"Seawater was pumped from a depth of 4 m to fill an airtight high grade stainless steel tank (200 L) designed for aerosol generation experiment. Sea ice samples were also introduced and melted in the tank for dedicated experiments. Water was dropped from the top of the tank as a plunging jet at a flow rate of 20 L min−1. The entrained air formed bubbles that, upon bursting, produced sea-spray aerosol, as reported in O'Dowd et al. (2015). Particle-free compressed air was blown into the tank headspace (120 L min⁻¹), which had outlet ports leading to samplers for the collection of filters and the subsequent off-line chemical characterization of the produced sea-spray. In particular nine sea-spray aerosol samples were collected for approximately 72h by a PM1 sampler (flow rate 40 lpm) equipped with pre-washed and pre-baked quartz-fiber filters (PALL, Ø= 47mm). Parallel bubble-bursting aerosol generation experiments with the same seawater and sea ice samples were carried out using a smaller glass tank (10 L) continuously flushed with particle-free air (11 L min⁻¹) (Schwier et al. 2015) and were dedicated to seaspray aerosol characterization using online mass spectrometers (HR-ToF-AMS and ATOFMS). The results from the bubble bursting experiments in the small tank are already reported in Dall'Osto et al. (2017)."

*3. In 3.3.1. Ambient aerosols from the Weddell Sea: Check the sample labelling in Figure 1. There are no information for sample A-0911.*

The correct labelling was indeed A-0901. We corrected the text.

*4. English expression is ambiguous in the manuscript. Please revise the whole of the paper to improve English expression.*

We have now removed the phrases with awkward syntax or with ambiguous expressions.

**References:**

Dall'Osto et al., Antarctic sea ice region as a source of biogenic organic nitrogen in aerosols, Scientific Reports, 7, 6047, doi:10.1038/s41598-017-06188-x, 2017.

Schwier et al. Primary marine aerosol emissions from the Mediterranean Sea during pre-bloom and oligotrophic conditions: correlations to seawater chlorophyll a from a mesocosm study. Atmos. Chem. Phys. 15, 7961–7976 (2015).